# SG2Loc: Sequential Visual Localization on 3D Scene Graphs

**Nicole Damblon** [1]  **Olga Vysotska** [1]  **Federico Tombari** [2,3]  **Marc Pollefeys** [1,4]  **Daniel Barath** [1,2]

## Abstract

Visual localization in complex indoor environments remains a critical challenge for robotics and AR applications. Sequential localization, where pose estimates are refined over time, is important for autonomous agents. However, traditional methods often require storing extensive image databases or point clouds, leading to significant overhead. This paper introduces a novel, lightweight approach to sequential visual localization using 3D scene graphs. Our method represents the environment with a compact scene graph, where nodes represent objects (with coarse meshes) and edges encode spatial relationships. For each image in the localization phase, we extract per-patch semantic features, predicting object identities. Localization is performed within a particle filter framework. Each particle, representing a camera pose, projects the coarse object meshes from the scene graph into the image, assigning object identities to patches based on visibility. The similarity of the per-patch features, in the input image, and object features from the scene graph determines the weight of a particle. Subsequent images are incorporated sequentially, refining the pose estimate. By leveraging a compact scene graph and efficient semantic matching, our method significantly reduces storage while maintaining performance on real-world datasets. The code is available at https://github.com/DmblnNicole/sg2loc.

## 1. Introduction

Visual localization is a fundamental capability in robotics and augmented reality. Accurate pose estimation (orienta-

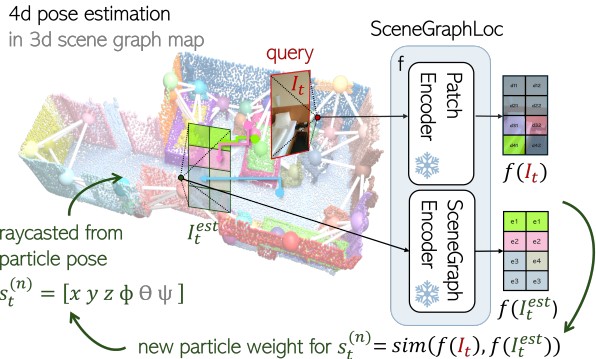

*Figure 1.* **Observation model** for a particle $\mathbf{s}_t^{(n)} = (x, y, z, \phi)$, matching object descriptors predicted by (Miao et al., 2024) from the query image $I_t$ to projected object labels from the semantically segmented coarse mesh. The particle weight reflects how well the observed and projected objects agree.

tion and position of an agent) enables autonomous navigation, scene understanding, and user-interaction tasks. Over the years, single-image localization and Simultaneous Localization and Mapping (SLAM)-based techniques have demonstrated remarkable progress (Arandjelovic et al., 2016; DeTone et al., 2018; Berton & Masone, 2025; Campos et al., 2021; Sattler et al., 2018; Murai et al., 2025), but they often demand large storage resources for image databases or 3D point clouds equipped with visual features. As environments grow both in scale and complexity, this burden becomes impractical for memory-constrained devices or applications with low bandwidth requirements.

Sequential localization strategies mitigate these challenges by incorporating temporal cues, refining pose estimates over multiple frames rather than treating each query image independently (Leutenegger et al., 2015; Campos et al., 2021; Maggio et al., 2022). Some works combine single-image pose estimation with SLAM to reduce drift and improve robustness (Lynen et al., 2020). Nonetheless, global pose alignment typically depends on retrieving a dense 3D model or querying large-scale maps. Such approaches can be prohibitive in extended deployments, especially during frequent map updates or transfers on resource-constrained platforms.

[1]ETH Zurich [2]Google [3]TU Munich [4]Microsoft. Correspondence to: Nicole Damblon <ndamblon@ai.ethz.ch>, Olga Vysotska <ovysotska@ethz.ch>, Federico Tombari <federico.tombari@tum.de>, Marc Pollefeys <marc.pollefeys@inf.ethz.ch>, Daniel Barath <daniel-bela.barath@inf.ethz.ch>.

*Proceedings of the 43rd International Conference on Machine Learning*, Seoul, South Korea. PMLR 306, 2026. Copyright 2026 by the author(s).

Scene graphs present a promising alternative. Storing the environment as a graph of objects and their spatial relationships offers compactness and a semantically rich representation. Early research on scene graphs primarily targeted scene understanding, reconstruction, and retrieval (Armeni et al., 2019; Johnson et al., 2015; Hudson & Manning, 2019). More recently, SceneGraphLoc (Miao et al., 2024) leveraged a scene graph to achieve cross-modal place recognition, highlighting their potential to reduce memory footprints while facilitating efficient localization.

Building on these insights, we introduce a novel method for sequential visual localization that combines a lightweight 3D scene graph with a particle filter. The proposed system relies on *coarse* object meshes, rather than dense point clouds, and *semantic* descriptors, significantly reducing storage overhead compared to the state of the art. We formulate sequential localization for indoor environments as a particle filtering problem, where each particle observes a portion of the scene and identifies object categories (e.g., *table* or *chair*). By recognizing objects, the system can evaluate whether the configuration of a particle (its pose) aligns with the input sequence by simply verifying whether the same objects appear in both the particle view and the query image (see Fig. 1). The main contributions of this paper are:

- A new 3D scene-graph-based framework for sequential visual localization that jointly models semantic cues, geometric and photometric constraints.
- A particle filter approach that leverages semantic object identities to refine camera poses iteratively, without requiring large image databases or point-cloud maps.
- As a technical contribution, we adapt SceneGraphLoc to work with image sequences (Sec. 4).

## 2. Related Work

Visual localization is a long-standing problem in computer vision and robotics, with roots in early works on Structure-from-Motion (Kruppa, 1913; Moravec, 1980; Fischler & Bolles, 1981). Modern approaches can be categorized into single-image localization, SLAM-based methods, and sequential localization methods. Our work falls into the last category, but leverages a novel scene graph representation, distinguishing it from prior art.

**Single-image localization.** Many recent methods rely on a two-stage approach: coarse localization (place recognition) followed by pose estimation. Coarse localization treats the problem as image retrieval, comparing a query image to a database of geo-tagged ones. Methods like NetVLAD (Arandjelovic et al., 2016), AP-GeM (Revaud et al., 2019a) and MegaLoc (Berton & Masone, 2025) provide global image descriptors for this purpose. Fine localization involves establishing 2D-3D matches between image features and

a 3D model (often a point cloud) of the scene, followed by pose estimation using RANSAC. Feature detectors and descriptors like SuperPoint (DeTone et al., 2018), R2D2 (Revaud et al., 2019b), and D2-Net (Dusmanu et al., 2019) are commonly used. While effective, these methods require storing large image databases or 3D point clouds equipped with visual features, leading to substantial storage demands. Alternative approaches, such as scene coordinate (Brachmann et al., 2017; Brachmann & Rother, 2018; Brachmann et al., 2023; Wang et al., 2024) and absolute pose regression (Walch et al., 2017; Kendall et al., 2015), directly predict 3D coordinates or camera poses from the image. However, these often struggle with complex, large-scale environments (Sattler et al., 2018; 2016b;a; Wang et al., 2024). Our work is fundamentally different, as it avoids storing image databases or point clouds by using a scene graph.

**Simultaneous localization and mapping** (SLAM) systems (Engel et al., 2014; Gao et al., 2018; Campos et al., 2021; Murai et al., 2025) provide local tracking of the trajectory of a camera. Popular examples include OpenVINS (Geneva et al., 2020), ORB-SLAM (Campos et al., 2021), DROID-SLAM (Teed & Deng, 2021), DSO (Engel et al., 2017), DPVO (Teed et al., 2023). While these systems excel at tracking relative motion, they are prone to accumulated drift over long trajectories and typically do not perform global localization (loop closure) without additional mechanisms. Some systems integrate inertial measurements (Lynen et al., 2020; Qin et al., 2018) for improved robustness. We build upon SLAM systems, leveraging their camera trajectory while providing lightweight global localization capabilities.

**Sequential localization.** Some approaches improve robustness by composing a local visual or visual-inertial tracker (Forster et al., 2014; Bloesch et al., 2015; Sattler et al., 2016b; Leutenegger et al., 2015; Qin et al., 2018), with a single-image global localizer (Middelberg et al., 2014; Lynen et al., 2020). Maplab (Schneider et al., 2018) provides a framework for visual-inertial benchmarking, while KFNet (Zhou et al., 2020) offers a learning-based alternative. Other methods incorporate temporal information by modeling image sequences, enabling joint localization across multiple frames. More recently, MASt3R-SLAM (Murai et al., 2025) and VGGT-SLAM (Maggio et al., 2026) use feed-forward 3D geometry for feature matching and mapping. In contrast, NeRF- and Gaussian Splat-based methods (Maggio et al., 2022; Adamkiewicz et al., 2022; Khatib et al., 2025; Meng et al., 2025) assume a pre-built dense reconstruction and localize by minimizing photometric error of rendered and observed views. While offering high-fidelity appearance matching, they are memory- and compute-intensive and remain highly sensitive to illumination and scene changes. We instead propose lightweight scene graphs with semantic and depth cues, providing compact maps and constraints in addition to photometric consistency.

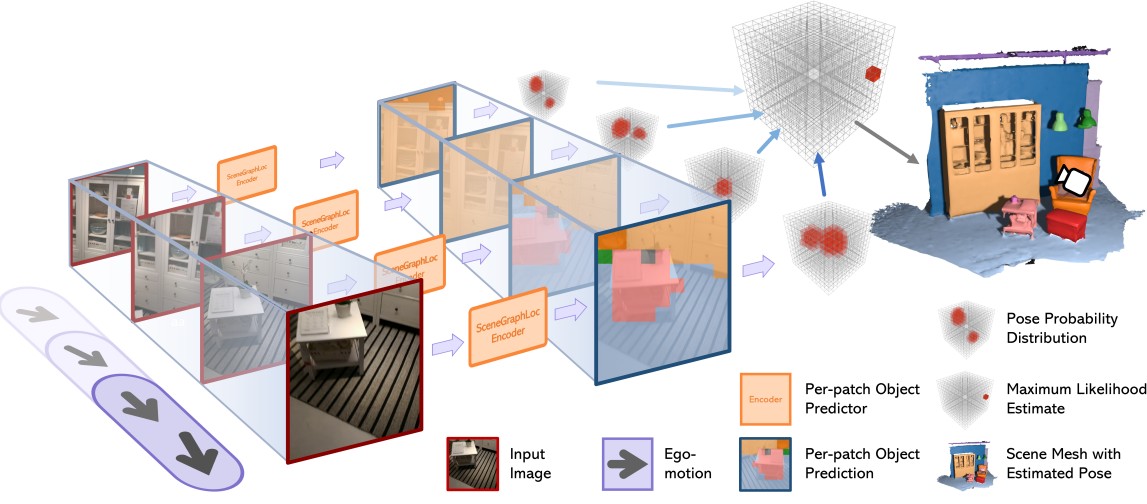

*Figure 2.* **Sequential localization pipeline.** Given the current image $I_t$ and its ego-motion, the pipeline updates the particle state while leveraging previously processed images $I_0, \ldots, I_{t-1}$. The current image is passed through the SceneGraphLoc (Miao et al., 2024) encoder to predict object labels for each image patch. These predictions inform a 4D probability distribution over the camera pose (3D position and rotation around the vertical axis), which is then integrated into the posterior distribution, refining the estimated camera pose as new images become available.

**3D scene graphs for localization and retrieval** have emerged as a powerful representation for capturing scene understanding, extending geometric representations with semantic and relational knowledge. Scene graphs first emerged as 2D representations of images, capturing objects and their relations for tasks such as image retrieval (Johnson et al., 2015) and visual question answering (Hudson & Manning, 2019). This idea was later lifted to 3D, where scene graphs are constructed from point clouds or RGB-D reconstructions to capture spatial and semantic structure (Armeni et al., 2019; Wald et al., 2020). The recent SceneGraphLoc (Miao et al., 2024) introduced the novel problem of cross-modal localization of a query image within a database of 3D scene graphs, demonstrating significant storage savings and faster query times compared to image-based methods. While not localization, it showed promise for place recognition. Our approach builds upon the strengths of sequential localization methods and the light-weight nature of scene graphs to provide a new localization direction.

## 3. Sequential Localization with Particle Filter

We aim to estimate a 4 degree-of-freedom camera pose $\mathbf{s}_t = (x, y, z, \phi)$, where $\phi$ denotes rotation around the known gravity axis, from a sequence of images $\{I_1, \ldots, I_T\}$. We maintain a set of $N$ particles $\mathbf{S}_t = \{\mathbf{s}_t^{(1)}, \ldots, \mathbf{s}_t^{(N)}\}$ that evolves over time using a particle filter. The filter integrates a *coarse*, labeled 3D mesh (derived from a 3D scene graph) with the pre-trained SceneGraphLoc model (Miao et al., 2024) to compute observation likelihoods. The particle filter pipeline is visualized in Fig. 2.

Note that the gravity and camera intrinsics are usually accessible from robot sensors, smartphones, and head-mounted devices. If this information is not readily available, methods such as GeoCalib (Veicht et al., 2024) can be used to estimate both the intrinsics and gravity direction.

**Scene representation.** We model the environment as a 3D scene graph $\mathcal{G} = (\mathcal{V}, \mathcal{E})$, where each vertex $v_i \in \mathcal{V}$ corresponds to an object instance $o_i$. Each object node $o_i$ is associated with:

- A compact embedding $\mathbf{e}_i \in \mathbb{R}^d$, obtained from the SceneGraphLoc object encoder (Miao et al., 2024), which combines multi-modal inputs (e.g., RGB images, point clouds, textual annotations, relationships to other objects).
- A coarse 3D mesh $\mathcal{M}_i$ that captures the approximate geometry of the object for raycasting. The mesh needs only be detailed enough for visibility and occlusion checks.[1]

Edges in $\mathcal{E}$ encode spatial relationships (e.g., adjacency) among objects, though our localization pipeline primarily relies on object embeddings $\{\mathbf{e}_i\}$ and coarse meshes $\{\mathcal{M}_i\}$. By storing these low-dimensional descriptors and approximate geometries instead of dense image databases or large point clouds, we greatly reduce the memory footprint while still enabling accurate pose estimation. We use existing methods to build scene graphs. For ScanNet, we use SceneGraphFusion (Wu et al., 2021), which incrementally predicts 3D scene graphs from RGB-D sequences. For

---

[1]On average, the 3RScan (Wald et al., 2019) meshes have 863 vertices per object and ScanNet (Dai et al., 2017) 11681 vertices per object in the experiments.

3RScan, scene graph annotations are provided. Our method is not tied to a specific scene graph construction method, it only requires object meshes and semantic embeddings. Any method that produces object-level 3D segmentations can be used, such as CLIO (Maggio et al., 2024), which builds scene graphs from open-vocabulary features in real time.

**Particle filter.** To approximate posterior $p(\mathbf{s}_t \mid I_{1:t})$, we maintain a set of weighted particles as

$$p(\mathbf{s}_t \mid I_{1:t}) \approx \sum_{n=1}^{N} w_t^{(n)} \, \delta\big(\mathbf{s}_t - \mathbf{s}_t^{(n)}\big), \qquad (1)$$

where $\delta$ is the Dirac delta function and $w_t^{(n)}$ denotes the normalized weight of the $n$-th particle at time $t$. At each timestep, the filter performs (i) a prediction step (motion model), (ii) an update step (observation model), and (iii) optional resampling, as described below in the next section.

### 3.1. Initialization

**Patch embeddings from SceneGraphLoc.** We subdivide each incoming image $I_t$ into a grid of $n \times m$ rectangular patches $\{p_{r,c}\}$, where $r \in [0, n)$ and $c \in [0, m)$. Following SceneGraphLoc (Miao et al., 2024), we use a $16 \times 9$ patch grid on 3RScan (144 patches) and a $32 \times 24$ grid on ScanNet (768 patches). We apply the SceneGraphLoc image encoder to obtain a set of patch embeddings $\{\hat{\mathbf{e}}_{r,c}\}$. Each $\hat{\mathbf{e}}_{r,c}$ is a semantic descriptor indicating which object (from the scene graph vocabulary) is most likely visible in the patch.

**Particle distribution.** We then uniformly distribute particles in a 3D bounding region $\Omega$ spanning four approximate camera heights $\{1.50, 1.60, 1.70, 1.80\}$ meters above the floor. Concretely, we partition $\Omega$ into uniform grid cells of size 0.2 meters, sample three poses per cell, and randomly assign each a yaw angle $\phi \in [-\pi, \pi]$. All particles receive equal initial weight, determined as follows

$$p(\mathbf{s}_0) = \frac{1}{|\Omega|} \text{ if } \mathbf{s}_0 \in \Omega \text{ else } 0 \qquad (2)$$

This broad initialization ensures coverage of plausible poses across the scene before the sequential estimation begins.

### 3.2. Prediction (Motion Model)

To propagate each particle $\mathbf{s}_t^{(n)}$ from timestep $t$ to $t + 1$, we apply the camera's ego-motion $\mathbf{t}_t = [\, t_{x,t}, \, t_{y,t}, \, t_{z,t}, \, t_{\phi,t} \,]$, augmented by i.i.d. Gaussian noise $\boldsymbol{\omega}_t = [\, \omega_{x,t}, \, \omega_{y,t}, \, \omega_{z,t}, \, \omega_{\phi,t} \,]$:

$$\mathbf{s}_{t+1}^{(n)} = \mathbf{s}_t^{(n)} \oplus \big( \mathbf{t}_t + \boldsymbol{\omega}_t \big). \qquad (3)$$

Here, $\oplus$ applies a 4-DoF transformation (3D trans./rot. around the gravity axis). We set $\sigma_{\text{trans}} = 0.05$ m and

$\sigma_{\text{rot}} = 0.05$ rad as the standard deviations for translation and yaw noise, respectively. This accounts for modeling uncertainty in the motion estimates while maintaining gravity alignment. Those parameters were set once and kept fixed across all experiments on both 3RScan and ScanNet. In practice, ego-motion is obtained by the employed SLAM system, for example (Teed & Deng, 2021).

### 3.3. Update (Observation Model)

After predicting the particle set $\mathbf{S}_{t+1}$, we incorporate the new image $I_{t+1}$ to refine particle weights.

**Raycasting.** For each particle pose $\mathbf{s}_{t+1}^{(n)}$, we project the coarse 3D meshes $\{\mathcal{M}_i\}$ from the scene graph into the image. This identifies which objects (nodes in $\mathcal{G}$) are visible in each of the $16 \times 9$ patches, assigning an object embedding $\mathbf{e}_i$ to each patch $\hat{p}_{r,c}$ for that particle.

**Patch similarity.** We then compare the patch embedding $\hat{\mathbf{e}}_{r,c}$ from the query image $I_{t+1}$ (Section 3.1) with the object embedding $\mathbf{e}_i$ assigned by raycasting. We compute the cosine similarity and retain it if the predicted object matches the one determined by raycasting. Summing these valid similarities across all patches and normalizing by the patch count yields a score per particle $s^{(n)} \in [0, 2]$, where the cosine similarity in $[-1, 1]$ is shifted to the positive range.

**Particle weighting.** We model the likelihood using a Gaussian centered on $s_{\max} = 2$, so that the likelihood is maximized when $s^{(n)} = s_{\max}$:

$$L\big(\mathbf{s}_{t+1}^{(n)}\big) = \exp\big(-(s_{\max} - s^{(n)})^2 / (2\,\sigma^2)\big),$$

with $\sigma = 0.2$. The unnormalized weight of each particle becomes:

$$\tilde{w}_{t+1}^{(n)} = w_t^{(n)} \cdot L(\mathbf{s}_{t+1}^{(n)}), \qquad (4)$$

which we normalize by $\sum_m \tilde{w}_{t+1}^{(m)}$ to obtain $w_{t+1}^{(n)}$. Particles whose similarity scores are closer to the maximum receive greater weight, driving the distribution toward accurate poses in subsequent steps.

### 3.4. Additional Supervision Signals

In addition to semantics, we use photometric and geometric cues to guide the particle filter.

**Color supervision.** As we are given an RGB sequence as input, we can leverage photometric losses to further improve our particle state quality measurements. Let us assume that our map representation is a coarse textured mesh. For each particle $\mathbf{s}_{t+1}^{(n)}$, we project the meshes $\{\mathcal{M}_i\}$ of the visible objects into the image plane. Note that this step does not require additional computations as the mesh has already been projected for semantic supervision. Now, instead of assigning object identities, we render an RGB image $I_p^{(n)}$

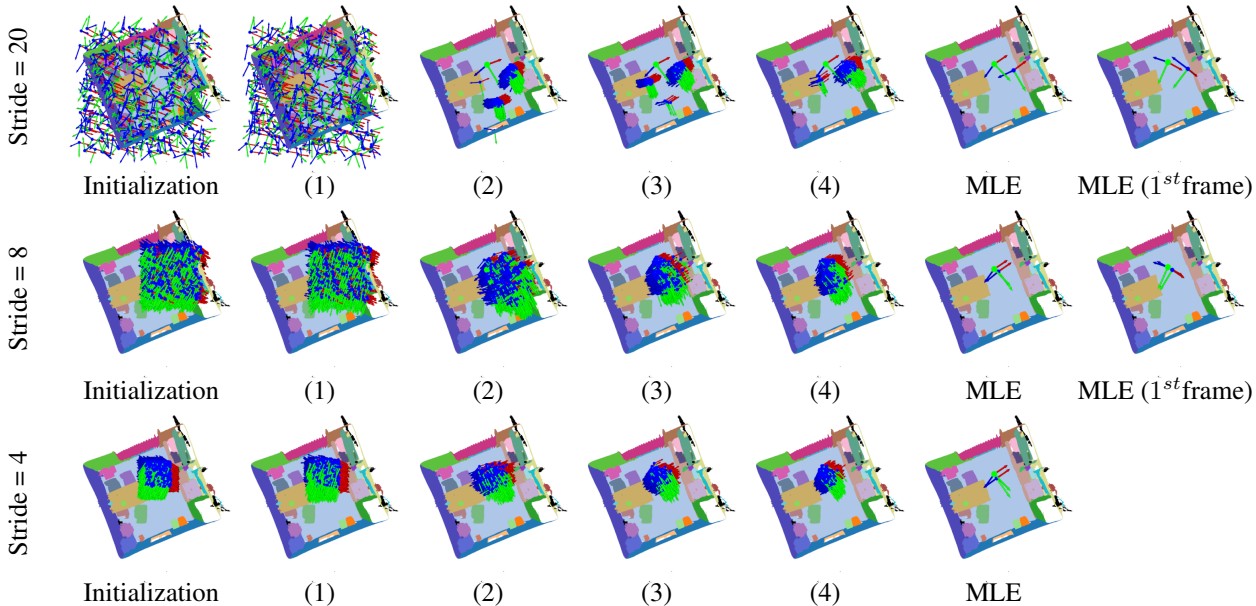

*Figure 3.* **Multi-round particle filter** on a 5-image sequence, first running with a stride of 20, 8, and finally 4 (the stride controls the rate of downsampling the images). The first column shows the initial random particle distribution, gradually narrowing the search space in subsequent rounds. Each next column represents particle updates after integrating the $n^{\text{th}}$ image (indicated below each plot). Then, the Maximum Likelihood Estimate (MLE) is shown in blue, and the GT pose in green, back-propagated to the $1^{st}$ frame for the next optimization round.

for the particle. Given the next image $I_{t+1}$ in the input sequence, we calculate the photometric score $s_i^{(n)}$ of the particle by calculating the structural similarity (SSIM) as $s_i^{(n)} = L_{\text{SSIM}}(I_{t+1}, I_p^{(n)})$, where $L_{\text{SSIM}}$ returns values between 0 and 1, and higher values indicate greater similarity.

**Depth supervision.** When depth information is available, either from an RGB-D sensor or a depth estimator applied to the image sequence, we can further incorporate this information to enhance accuracy. We achieve this by adding a depth-based score. This allows us to guide the particles also through the geometric structure of the scene.

*Depth map projection.* For each particle $\mathbf{s}_{t+1}^{(n)}$, we project the coarse 3D meshes of the visible objects into the image plane. Now, instead of assigning object identities, we render a depth map $D_p^{(n)}$ for the particle. This depth represents the distance from the camera plane to the projected mesh surfaces, according to the pose of the particle. We use the same resolution for $D_p^{(n)}$ as the input image (or the downsampled version, as described in Sec. 3.5). Let $D_I$ be the depth map corresponding to the input image $I_{t+1}$. If $I_{t+1}$ is an RGB-D image, $D_I$ is directly obtained from the sensor. If only RGB data is available, $D_I$ can be estimated using a monocular or stereo depth estimation method.

*Depth score calculation.* We compute the depth score $s_d^{(n)}$ for particle $\mathbf{s}_{t+1}^{(n)}$ by calculating the L1 difference of the projected depth map $D_p^{(n)}$ and the input $D_I$. The score is

calculated as follows:

$$s_d^{(n)} = 1 - \frac{1}{R \cdot C} \sum_{r=1}^{R} \sum_{c=1}^{C} |D_p^{(n)}(r,c) - D_I(r,c)|, \quad (5)$$

where $R$ and $C$ are the number of rows and columns of the depth maps, respectively, and $D_p^{(n)}(r,c)$ and $D_I(r,c)$ are the depth values at pixel $(r,c)$ in the projected and input depth maps, respectively. The score is designed such that 1 is best and lower values are worse.

**Combined score.** The final weight is then computed by combining the semantic similarity score $s^{(n)}$ (from Sec. 3.3), the depth $s_d^{(n)}$ and color scores $s_i^{(n)}$ as:

$$L\big(\mathbf{s}_{t+1}^{(n)}\big) = \exp\!\Big(-\frac{(s_{\max}-(\lambda_0 s^{(n)}+\lambda_1 s_d^{(n)}+\lambda_2 s_i^{(n)}))^2}{2\,\sigma^2}\Big), \quad (6)$$

where $\lambda_0$, $\lambda_1$, $\lambda_2$ balance the contribution of the semantic, depth and color scores, and $s_{\max}$ is the maximum value the weighted sum can attain. The weights are set empirically. For our experiments, all scores contribute equally ($\lambda_0 = \lambda_1 = \lambda_2 = 1$) and are kept fixed across all experiments. The unnormalized weight becomes:

$$\tilde{w}_{t+1}^{(n)} = w_t^{(n)} \cdot L\big(\mathbf{s}_{t+1}^{(n)}\big), \quad (7)$$

We normalize these weights as before: $w_{t+1}^{(n)} = \tilde{w}_{t+1}^{(n)} / \sum_m \tilde{w}_{t+1}^{(m)}$. This combined scoring mechanism leverages semantic, geometric and appearance information, leading to more robust and accurate localization, especially

where the semantic information alone might be ambiguous. The depth and color scores provide additional constraints based on geometric and photometric consistency, helping to disambiguate poses with similar semantic projections.

### 3.5. Coarse-to-Fine Optimization

We refine the pose estimate over multiple rounds, gradually increasing raycasting resolution and narrowing the search region. We observed that this process speeds up the localization and leads to higher accuracy. In the first round, we downsample the input image with a stride of 20 pixels, enabling a coarse but efficient search over the entire scene. Particles are initialized uniformly within the full environment bounds, allowing a rapid, coarse localization. Next, we apply an 8-pixel stride, restricting the particle initialization to a $1.6^3$ m region centered around the maximum likelihood estimate (MLE) pose from the previous round. This second pass refines localization by focusing on a smaller region. Finally, we downsample the image with stride 4 and use an even tighter bounding box of $0.8^3$ m around the new MLE pose. This final pass incorporates high-resolution visibility checks and yields the most precise pose estimate.

**Adaptive resampling.** Following each update, we dynamically adjust the particle count using an adaptive scheme based on KLD-sampling (Fox, 2001). Letting $\epsilon$ and $\delta$ be bounds on the Kullback–Leibler divergence and its confidence level, respectively, we compute

$$n = \frac{1}{2\epsilon} \left( 1 - \frac{2}{9(k-1)} + \sqrt{\frac{2}{9(k-1)}} \, z_{1-\delta} \right)^3, \quad (8)$$

where $k$ is the bin number in the state histogram used for divergence estimation, and $z_{1-\delta}$ is the $(1-\delta)$-quantile of the normal distribution. We then apply stratified resampling to draw $n$ new particles, ensuring that the MLE distribution remains an accurate approximation of the true posterior. An example visualization is shown in Fig. 3.

### 3.6. Pose Refinement with PnP

To refine the final particle filter pose, we render six synthetic views from the mesh. One from the MLE pose estimated in the previous steps and five sampled within a range of $\pm45°$ yaw around it. We match each view with the query image using RoMav2 (Edstedt et al., 2025) and establish 2D-3D correspondences through a ray-mesh intersection. Pose refinement uses RANSAC-based PnP from PoseLib (Larsson & contributors, 2020). For each sequence, we apply this process per frame by backpropagating the MLE pose (obtained from the last frame) to earlier images. We select the pose with the highest inlier count as the final estimate.

## 4. Sequential Scene Retrieval

The particle filter in Sec. 3 estimates a camera pose within a *known* scene graph. In multi-scene deployments, one first needs to identify *which* scene the agent is in. The two stages are independent: the particle filter works without retrieval (by uniformly sampling the space), and retrieval can stand alone as a place-recognition step. For completeness, we propose a sequential extension of (Miao et al., 2024) for this complementary scene-identification step.

SceneGraphLoc was originally designed to select the correct scene graph from a database $\{\mathcal{G}_i\}$ given a single query image. We adapt it to leverage an entire input sequence instead. SceneGraphLoc (Miao et al., 2024) was originally designed to select the correct scene (represented as a 3D scene graph) from a set of candidate scene graphs $\{\mathcal{G}_i\}$ given a query image as input. In this section, we adapt SceneGraphLoc to leverage not just one image but the entire input sequence when retrieving the current scene from the database.

For each image $I_t$ in the sequence, we extract a set of image patches $\mathcal{Q}_t$. For each patch $q \in \mathcal{Q}_t$, we compute its embedding $e_q$ using the pre-trained encoder. Similar to (Miao et al., 2024), we then compare $e_q$ with the embeddings $e_v$ of nodes $v \in \mathcal{V}_i$ in each scene graph $\mathcal{G}_i$ in our database using a similarity metric as $\text{similarity}(q, v) = \cos(e_q, e_v)$. We assign a score to each scene graph $\mathcal{G}_i$ based on the similarity scores of its nodes to the image patches as: $\text{score}_t(\mathcal{G}_i, I_t) = \frac{1}{|\mathcal{Q}_t|} \sum_{q \in \mathcal{Q}_t} \max_{v \in \mathcal{V}_i} \text{similarity}(q, v)$. To incorporate information from the sequence, we aggregate the scores from all images as $\text{score}(\mathcal{G}_i) = \sum_{t=1}^{T} \text{score}_t(\mathcal{G}_i, I_t)$. The scene graph with the highest final score is selected as the correct match for the sequence. This process allows us to better find the correct scene in a database of maps.

## 5. Experiments

**Datasets.** The *3RScan* dataset (Wald et al., 2019) contains 1,335 annotated indoor scenes across 432 spaces, with 1,178 scenes (385 rooms) for training and 157 (47 rooms) for validation. Each scene is represented by a semantically annotated 3D point cloud, with multiple captures over months to reflect environmental changes. For 3RScan, scene graph annotations are provided with the dataset. Since the test set lacks scene graph annotations, we follow (Miao et al., 2024) and reorganize the validation set into 34 scenes (17 rooms) for validation and 123 scenes (30 rooms) for testing. To ensure realistic evaluation, query sequences are localized against maps from different temporal states. We evaluate on 3RScan using the 30 test scenes with sequence lengths of 5, 10, and 25 frames, corresponding to 1865, 916, and 349 sequences (9325, 9160, and 8725 total images), respectively. We note that 3RScan is explicitly designed to evaluate localization under significant scene changes, including object

*Table 1.* **Pose recalls and median errors** on 3RScan and ScanNet. We report recalls at thresholds: position R@0.25m, rotation R@2° and the joint recall R@0.25m, 2°, together with the median position (m) and rotation error (°) for sequences of length 5, 10 and 25. Recall measures the fraction of errors that fall below those thresholds, higher is better. ACE and ACE+GS-CPR are shown faded on ScanNet since the ACE encoder was trained on parts of this dataset.

| | Method | 5 frames | | | | | 10 frames | | | | | 25 frames | | | | |
|---|---|---|---|---|---|---|---|---|---|---|---|---|---|---|---|---|
| | | Recall ↑ | | | Med. err. ↓ | | Recall ↑ | | | Med. err. ↓ | | Recall ↑ | | | Med. err. ↓ | |
| | | m | ° | m,° | m | ° | m | ° | m,° | m | ° | m | ° | m,° | m | ° |
| **3RScan** | HLoc | 0.54 | 0.29 | 0.29 | 0.19 | 6.65 | 0.60 | 0.34 | 0.34 | 0.14 | 4.30 | 0.70 | 0.43 | 0.43 | 0.08 | 2.69 |
| | MeshLoc | 0.55 | 0.29 | 0.29 | 0.19 | 5.91 | 0.62 | 0.34 | 0.33 | 0.14 | 3.93 | 0.69 | 0.41 | 0.41 | 0.08 | 2.61 |
| | Loc-NeRF | 0.03 | 0.05 | 0.00 | 1.19 | 21.63 | 0.06 | 0.04 | 0.01 | 1.02 | 22.00 | 0.11 | 0.13 | 0.04 | 0.79 | 14.08 |
| | ACE | 0.26 | 0.11 | 0.11 | 1.66 | 42.38 | 0.33 | 0.14 | 0.14 | 0.93 | 27.04 | 0.45 | 0.21 | 0.21 | 0.39 | 10.31 |
| | ACE + GS-CPR | 0.33 | 0.14 | 0.14 | 1.61 | 50.13 | 0.42 | 0.19 | 0.19 | 0.75 | 20.88 | 0.54 | 0.27 | 0.27 | 0.18 | 5.93 |
| | **SG2Loc (Ours)** | **0.58** | **0.39** | **0.34** | **0.15** | **2.92** | **0.66** | **0.42** | **0.39** | **0.11** | **2.55** | **0.75** | **0.46** | **0.44** | **0.07** | **2.21** |
| **ScanNet** | HLoc | **0.94** | 0.53 | 0.53 | **0.06** | **1.85** | **0.96** | 0.57 | 0.57 | **0.06** | **1.74** | **0.99** | 0.58 | 0.58 | **0.06** | **1.56** |
| | MeshLoc | 0.93 | **0.55** | **0.54** | **0.06** | 1.83 | 0.95 | 0.55 | 0.55 | **0.06** | 1.83 | 0.97 | 0.58 | 0.58 | **0.06** | 1.70 |
| | Loc-NeRF | 0.03 | 0.08 | 0.00 | 1.09 | 17.60 | 0.05 | 0.08 | 0.01 | 1.11 | 15.10 | 0.21 | 0.14 | 0.08 | 0.77 | 9.02 |
| | ACE | 0.91 | 0.50 | 0.50 | 0.08 | 1.99 | 0.94 | 0.52 | 0.51 | 0.08 | 1.93 | 0.94 | 0.60 | 0.59 | 0.07 | 1.70 |
| | ACE + GS-CPR | 0.92 | 0.55 | 0.55 | 0.06 | 1.84 | 0.93 | 0.58 | 0.58 | 0.06 | 1.78 | 0.92 | 0.63 | 0.63 | 0.06 | 1.53 |
| | **SG2Loc (Ours)** | 0.87 | 0.49 | 0.49 | 0.07 | 2.06 | 0.94 | 0.52 | 0.52 | 0.07 | 1.90 | 0.94 | **0.60** | **0.60** | **0.06** | 1.79 |

rearrangements, removal, and occlusions. SG2Loc is evaluated on every frame of this dataset in a cross-temporal setting, where query sequences are matched against maps captured some time apart. For evaluation without GT scene graphs, we also use *ScanNet* (Dai et al., 2017). Starting from the split of (Miao et al., 2024), we filter for scan pairs with both a reference (_00) and rescan (_01), and for which SceneGraphFusion (Wu et al., 2021) generation succeeded, yielding 48 pairs. We sample one image every 25 frames because frame rate is high (like SceneGraphLoc), resulting in 683, 310, and 95 sequences (3,415, 3,100, and 2,375 query frames) for sequence lengths 5, 10, and 25, respectively.

**Baselines.** We compare with HLoc (Sarlin et al., 2019), MeshLoc (Panek et al., 2022), Loc-NeRF (Maggio et al., 2022), ACE (Brachmann et al., 2023) and ACE poses refined with GS-CPR (Liu et al., 2025). HLoc is a hierarchical method combining image retrieval (10 nearest neighbors) with pose estimation from 2D-3D correspondences, requiring a large image database. MeshLoc localizes via a 3D mesh, storing a depth map per image generated from the mesh. We use MegaLoc (Berton & Masone, 2025) for image retrieval, SuperPoint (DeTone et al., 2018) and SuperGlue (Sarlin et al., 2020) for local feature matching. ACE is a scene coordinate regression network trained per scene. GS-CPR refines ACE by using rendered depth from Gaussian Splats and MASt3R correspondences. For sequential evaluation, we run HLoc, MeshLoc, ACE, and ACE+GS-CPR per image and select the pose with the most RANSAC inliers as the sequence result. Loc-NeRF follows a similar filtering approach as we do, using only photometric error. We replaced the original NeRF map with a Gaussian Splat for faster localization and refer to this adapted version as Loc-NeRF throughout. HLoc and MeshLoc both require substantial storage, limiting scalability.

**Metrics.** We evaluate the performance of our method and baselines using standard metrics for visual localization: median position (in meters), rotation errors (in degrees) and recall. Additionally, we report the storage requirements of each method, including the size of the database and any additional structures needed for localization. The rotation error is calculated as the geodesic distance between the estimated rotation matrix $\hat{R}$ and the ground truth rotation matrix $R_{gt}$ on the SO(3) manifold. It is calculated as follows: $\epsilon_R = \arccos((1/2)(\operatorname{tr}(R_{gt}^T \hat{R}) - 1))$.

**Sequential visual localization.** Table 1 shows that our method achieves the best joint recalls for all frame lengths on 3RScan. On ScanNet, it performs best for 25 frames for joint recall and position median. We obtain the best median errors and recall (all metrics) on 3RScan. HLoc and MeshLoc perform well across all sequence lengths and yield very similar results. ACE performs worse than HLoc and other baselines on 3RScan, but achieves high accuracy on ScanNet. We excluded the scenes that the ACE encoder has been trained on from evaluation. Our method is roughly as good as ACE on ScanNet, especially for the longer seqeunces. The GS-CPR refinement of ACE poses (ACE+GS-CPR) improves ACE. All methods benefit from longer sequences, which consistently improve performance.

Although all methods are built from the same input images, the resulting map representations differ substantially in size. SG2Loc achieves the best results on 3RScan across all sequence lengths and performs on par with HLoc, MeshLoc, and ACE on ScanNet, while requiring roughly an order of magnitude less storage than HLoc and two orders of magnitude less than MeshLoc on 3RScan, and two orders of magnitude less than both on ScanNet (Table 2). This significant reduction in storage makes SG2Loc well-suited for

Table 2. **Average storage** per scene in MB.

| Method | 3RScan | ScanNet |
|---|---|---|
| HLoc | 57.6 | 2291.3 |
| MeshLoc | 701.3 | 10591.0 |
| Loc-NeRF | 143.9 | 66.6 |
| ACE | **4.2** | **4.2** |
| ACE+GS-CPR | 82.6 | 115.1 |
| **SG2Loc (Ours)** | 9.8 | 28.2 |

Table 3. **Average runtime on 3RScan** per scene (offline, seconds) for frame integration and final optimization by PnP (runs once). The localization time is averaged over sequences and accounts for the *per-frame* integration.

| Method | Mapping | Localization | Final Opt. |
|---|---|---|---|
| HLoc | 333.6 | 0.32 | – |
| MeshLoc | 337.9 | 0.93 | – |
| Loc-NeRF | 1490.0 | 7.50 | – |
| ACE | 134.2 | **0.08** | – |
| ACE+GS-CPR | 924.0 | 0.28 | – |
| **SG2Loc (Ours)** | **1.6** | 2.90 | 4.2 |

on-device localization, where efficient map storage and low-bandwidth transmission are critical constraints. Our method uses roughly 2500 particles on 3RScan and 3200 particles on ScanNet for the results reported in Table 1. Raycasting for all particles is run fully in parallel on GPU, which keeps runtime manageable even with larger particle sets.

The mapping and per-frame processing times (in seconds) are reported in Table 3. For the proposed method, the mapping time includes constructing a BVH-tree for raytracing with particles and computing object embeddings for the scene graph. We assume the scene graph and segmented mesh are given, they do not count into the mapping time reported in 3. For HLoc, it includes extracting image embeddings and 2D-3D correspondences for the database (from the mesh), while for MeshLoc, it involves computing both image embeddings and depth maps. Loc-NeRF, ACE and GS-CPR require training a separate map representation for each scene. The localization for SG2Loc consists of state transitions, particle updates (embeddings are computed per query image and raycasting is performed from all particles), and resampling steps, before post-processing with PnP. The localization time represents the average time required to estimate the camera pose for a single frame. The proposed method requires substantially less computation during mapping than the baselines. At inference, it runs at an average of 2.9 s per frame, followed by a one-time pose optimization step, which is suitable for sequential localization.

**Keyframing.** In practice, localization need not be performed on every frame, as high frame rates introduce re-

dundancy. To simulate a realistic scenario, we integrate frames at a rate matched to our runtime (Table 5), where the agent moves in real time and incoming frames must be processed sequentially. To reflect online deployment, both HLoc and our method are evaluated per keyframe, rather than selecting the maximum-inlier pose across a sequence. We report results over 650 keyframes spaced apart by our per-frame runtime. Our method significantly outperforms HLoc on the same inputs, demonstrating its practicality for robot localization on resource-constrained devices.

**Sequential scene retrieval.** In this section, we evaluate the extension proposed in Section 4, which enables Scene-GraphLoc (Miao et al., 2024) to operate in a sequential setting. We compare this extension to the original method, which performs localization using only the first image of the sequence. Additionally, we benchmark against state-of-the-art image retrieval methods, AnyLoc (Keetha et al., 2023) and CVNet (Lee et al., 2022). We follow the evaluation protocol used in (Miao et al., 2024), where, given a query image (or sequence), we retrieve the top-$k$ scenes from a candidate set of either 10 or 50 scenes. Query images are captured at a different time step than the reference map. We report recall at ranks 1, 3, and 5, measuring how often each method retrieves the correct scene among its top-$k$ predictions ($k \in \{1, 3, 5\}$). Additionally, we provide localization time in milliseconds and storage requirements for retrieval. For our method, storage corresponds to the scene graph embeddings, whereas for CVNet and AnyLoc, it reflects the database of image embeddings.

The results are presented in Table 4. As expected, using longer sequences for localization substantially and consistently improves retrieval recall in both the 10-scene and 50-scene settings while introducing only a minimal increase in runtime. These findings demonstrate that the sequential extension of SceneGraphLoc is effective and serves as a valuable complement to the proposed localization approach.

**Ablation studies** are conducted on the first 4 scenes of the 3RScan (Wald et al., 2019) dataset, comprising 335 sequences of 5 images. The results are presented in Table 6. The first 3 ablations evaluate components of our method relative to *SG2Loc w/ semantic*. The *Max. resolution* (stride = 1) setting removes downsampling in the final pass of multi-round optimization, using the highest resolution for raycasting. As shown in Table 6, this configuration achieves similar accuracy to *SG2Loc w/ semantic*, while achieving much lower computational cost. The *Uniform sampling* setting initializes particles on a uniform 3D grid with a resolution of 0.2m, assigning 3 random poses per grid cell. This approach results in significantly lower accuracy compared to *SG2Loc w/ semantic* with the proposed init. strategy. The *w/o adaptive resampling* configuration replaces the adaptive resampling with a fixed particle count. While this achieves similar med. errors, it significantly re-

*Table 4.* **Cross-modal sequential scene retrieval** on the 3RScan dataset (Wald et al., 2019), where the goal is to identify the correct scene from a set of 10 or 50 candidates. We evaluate SceneGraphLoc (Miao et al., 2024) on single frames and our proposed extension, which enables (Miao et al., 2024) to process sequences. We also show results of standard techniques (copied from (Miao et al., 2024)) CVNet (Lee et al., 2022) and AnyLoc (Keetha et al., 2023). Tests are run on sequences of length 5, 10, 25, and 50. We report scene retrieval recall in the temporal setting ($R^t$) at 1, 3, and 5 indicating whether the correct scene appears among the top-$k$ predictions, along with inference time (ms) and storage requirements for the map, demonstrating the efficiency of our approach.

| | 10 scenes | | | | 50 scenes | | | | Storage |
| # of frames | $R^t$@1 | $R^t$@3 | $R^t$@5 | Time (ms) | $R^t$@1 | $R^t$@3 | $R^t$@5 | Time (ms) | MB |
|---|---|---|---|---|---|---|---|---|---|
| 1 ((Miao et al., 2024)) | 0.82 | 0.94 | **0.98** | **0.3** | 0.69 | 0.79 | 0.84 | **1.5** | **5.4** |
| 5 | 0.84 | 0.95 | **0.98** | 1.5 | 0.72 | 0.84 | 0.88 | 7.5 | **5.4** |
| 10 | 0.86 | 0.95 | **0.98** | 3.0 | 0.75 | 0.85 | 0.89 | 15.0 | **5.4** |
| 25 | 0.88 | **0.96** | **0.98** | 7.5 | 0.78 | 0.87 | 0.91 | 37.5 | **5.4** |
| 50 | **0.89** | **0.96** | **0.98** | 15.0 | **0.81** | **0.89** | **0.92** | 75.0 | **5.4** |
| CVNet (Lee et al., 2022) | 0.79 | 0.91 | 0.95 | 60.0 | 0.67 | 0.77 | 0.82 | 311.1 | 239.1 |
| AnyLoc (Keetha et al., 2023) | 0.88 | 0.95 | **0.98** | 1826.4 | **0.81** | 0.87 | 0.90 | 1451.1 | 5720.3 |

*Table 5.* **Keyframing on 3RScan.** We report avg. and median pos. (m) and rot. error (°), and recall at (25cm, 2°).

| | HLoc | Ours w/ keyframing |
|---|---|---|
| Mean pos. (m) ↓ | 16.55 | **1.13** |
| Mean rot. (°) ↓ | 57.02 | **12.39** |
| Med. pos. (m) ↓ | 0.63 | **0.24** |
| Med. rot. (°) ↓ | 24.10 | **4.26** |
| Pos. R@0.25m ↑ | 0.41 | **0.51** |
| Rot. R@2° ↑ | 0.20 | **0.31** |
| R@25cm, 2° ↑ | 0.20 | **0.28** |

*Table 6.* **Ablation study on 3RScan.** We report recall at (10cm, 5°) and (25cm, 10°) for 5-frame sequences.

| Method | R@10cm, 5° | R@25cm, 10° |
|---|---|---|
| Max. raycast resolution (3.5) | 0.07 | 0.33 |
| Uniform sampling (3.1) | 0.10 | 0.30 |
| w/o adaptive resampling (3.5) | 0.11 | 0.30 |
| SG2Loc w/ sem. (3.4) | 0.11 | 0.35 |
| SG2Loc w/ sem.+depth (3.4) | 0.20 | 0.54 |
| SG2Loc w/ sem.+depth+RGB (3.4) | 0.35 | 0.61 |
| **SG2Loc (Ours)** (3.6) | **0.50** | **0.65** |

duces recall, showing that dynamic particle count increases accuracy. The ablations *SG2Loc w/ semantic*, *w/ semantic+depth*, *w/ semantic+depth+RGB*, explore the impact of supervision signals without PnP. Each additional signal improves performance. The best variant (*SG2Loc w/ semantic+depth+RGB*) initializes the PnP refinement, and our full SG2Loc method (last row), yields the best overall results.

**Limitations.** Although the proposed method achieves substantial storage savings and competitive localization accuracy compared to the state-of-the-art HLoc and MeshLoc, it incurs higher computational cost during per-frame integration. We believe that this overhead can be significantly reduced through code optimizations. Moreover, the keyframing experiment highlights that SG2Loc can already be used in time-sensitive applications by processing only keyframes.

# 6. Conclusion

We present a lightweight approach to sequential visual localization using 3D scene graphs and a particle filter, avoiding the need for large image databases or dense point clouds. By leveraging semantic object descriptors and coarse meshes, our method efficiently refines pose estimates over time while significantly reducing storage requirements. Experiments show that SG2Loc achieves competitive accuracy with far lower storage overhead than existing methods. The proposed coarse-to-fine optimization balances efficiency and precision, making the approach practical for resource-constrained applications. SG2Loc achieves performance similar to storage-intensive baselines, sometimes it is better in accuracy, sometimes slightly worse. Future work will focus on enhancing feature representations and runtime.

## Acknowledgements

This work was supported by RobotX. Nicole Damblon is supported by the ETH AI Center doctoral fellowship. We thank both programs for their support.

## Impact Statement

This paper presents work whose goal is to advance the field of computer vision and robotics. There are many potential societal consequences of our work, none of which we feel must be specifically highlighted here.

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

## A. Additional Ablations on Supervision Signals

To understand the individual contribution of each supervision signal, we run the particle filter using only a single score at a time (semantic, depth, or RGB). Table 7 reports the resulting localization accuracy and pose recall on 3RScan (Wald et al., 2019). *SG2Loc w/ semantic* achieves the lowest median position error and consistently good performance in all metrics. *SG2Loc w/ depth* has the lowest median rotation error. In contrast, *SG2Loc w/ RGB* performs worse in terms of position and rotation accuracy, but interestingly achieves the highest recall at the strict threshold $(10\text{cm}, 5°)$. These results suggest that all three supervision signals contribute useful information to the final localization performance.

*Table 7.* **Ablations on supervision signals on 3RScan**. We report the average and median position error (in meters) and average and median rotation error (in degrees) for sequences of 5 frames, where lower values indicate better performance. We also measure recall at thresholds $(10\text{cm}, 5°)$ and $(25\text{cm}, 10°)$, where higher values indicate better performance.

| Method | Mean | | Median | | Recall | |
|---|---|---|---|---|---|---|
| | Pos. (m) | Rot. (°) | Pos. (m) | Rot. (°) | R@10cm, 5° | R@25cm, 10° |
| SG2Loc w/ semantic | **1.20** | **21.44** | **0.66** | 8.77 | 0.08 | 0.26 |
| SG2Loc w/ depth | 1.44 | 39.01 | 0.96 | **6.42** | 0.09 | **0.32** |
| SG2Loc w/ RGB | 1.79 | 44.78 | 1.51 | 17.37 | **0.14** | 0.20 |

## B. Qualitative Results

Figure 4 shows a failure case of the particle filter on a 5-frame sequence from the 3RScan dataset (Wald et al., 2019). Although the particle distribution narrows over time, the final estimate still results in a coarse localization with a pos. error of 1.06 meters and a rot. error of 6.6 degrees. This example highlights a limitation of our method. The input views in the query sequence look very similar (Figure 5), with little change in perspective. In such cases, the information in the image sequence is not discriminative enough to resolve pose ambiguities. As a result, the method struggles to converge to a precise pose and only returns a coarse estimate.

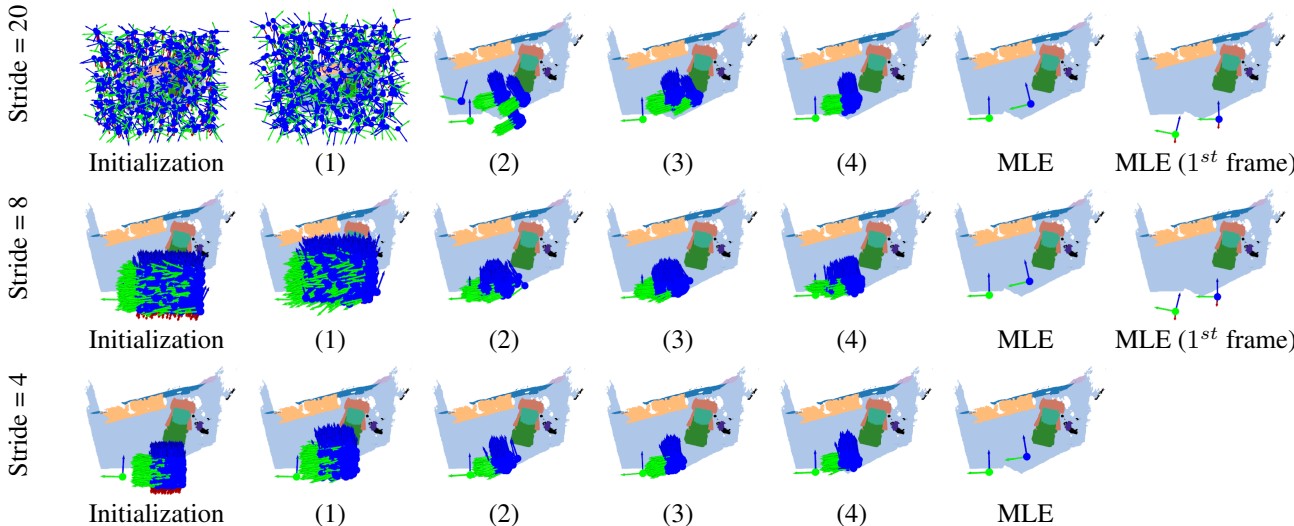

*Figure 4.* **A failure case** of our coarse-to-fine particle filter on a 5-image sequence, first running with a stride of 20, 8, and finally 4 (the stride controls the rate of downsampling the images). The first column shows the initial random particle distribution, gradually narrowing the search space in subsequent rounds. Each next column represents particle updates after integrating the $n^{\text{th}}$ image (indicated below each plot). Then, the Maximum Likelihood Estimate (MLE) is shown in blue, and the GT pose in green, back-propagated to the $1^{st}$ frame for the next optimization round. The position error for this example is 1.06 meter and rotation error 6.60°. This failure is potentially caused by the uninformative input views (all looking very similar) visualized in Fig. 5.

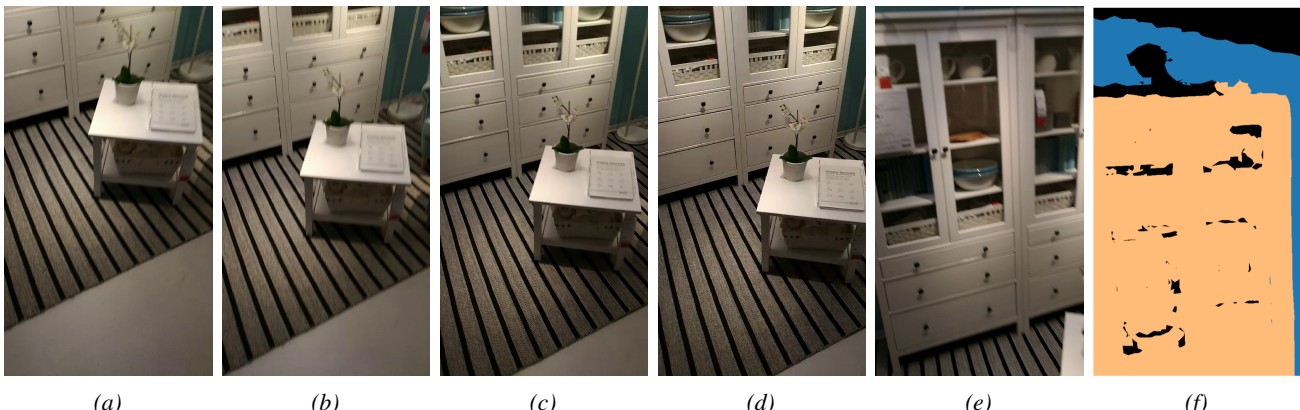

|     (a)     |     (b)     |     (c)     |     (d)     |     (e)     |     (f)     |

*Figure 5.* **Query images and MLE view.** The five query images (a)-(e) used in Figure 4 for the course-to-fine particle filter on the 3RScan dataset (Wald et al., 2019). After the fifth image (e), we retrieve the Maximum Likelihood Estimate (MLE) pose. Image (f) shows the segmented 3D mesh projected into a virtual camera located at the MLE pose. Colors denote object instances.

*Table 8.* Results on ScanNet (all frames not only last frame) using DROID-SLAM poses

| Method | Med. Pos. ↓ | Med. Rot. ↓ | R@0.25m ↑ | R@2° ↑ | R@0.25m,2° ↑ |
|---|---|---|---|---|---|
| SG2Loc (ours) | 0.089m | 2.41° | 81.5% | 40.5% | 40.4% |
| SG2Loc (ours) /w SLAM poses | 0.086m | 2.37° | 81.1% | 40.3% | 40.2% |

## C. Experiment using DROID-SLAM poses

We additionally report results using DROID-SLAM (Teed & Deng, 2021) for the relative motion on ScanNet (sequence length 5) in table 8. The performance is comparable, showing that SG2Loc is robust to moderate drift in the motion estimate. This are the results from the whole sequence. We do not pick the maximum inlier here.

## D. Large-scale environments

We conducted an additional experiment simulating a larger-scale environment. Since the available datasets do not provide large-scale, object-annotated environments (all scenes are apartment-sized), we merged every 3 ScanNet scenes into a single environment, initializing particles across the entire space and allowing the filter to automatically determine the correct scene. We report both the retrieval accuracy ($R^t@1$, i.e., fraction of cases where the correct room was identified) and the final pose accuracy. The results for 50-frame sequences, shown below, demonstrate that our method can successfully localize in this scenario. It achieves slightly lower retrieval performance compared to the proposed sequential SceneGraphLoc and the original SceneGraphLoc, while still producing accurate poses. The lower recall could potentially be improved by increasing the number of initial particles. Given its efficiency, running the sequential SceneGraphLoc variant for the coarse localization stage remains preferable. As per standard practice in localization (Sarlin et al., 2019), SG2Loc can provide an accurate pose after the room has been identified by the proposed sequential SceneGraphLoc.

*Table 9.* Localization results **large-scale environment** on ScanNet.

| Method | $R^t@1$ | Mean Pos. (m) ↓ | Mean Rot. (°) ↓ | Med. Pos. (m) ↓ | Med. Rot. (°) ↓ | R@10cm,5° ↑ | R@25cm,10° ↑ |
|---|---|---|---|---|---|---|---|
| SGL (Miao et al., 2024) | 92.4 | – | – | – | – | – | – |
| Seq. SGL (Ours) | 95.1 | – | – | – | – | – | – |
| SG2Loc (Ours) | 85.7 | 0.24 | 7.21 | 0.16 | 2.25 | 0.33 | 0.80 |

