# OpenReview forum: "SG2Loc: Sequential Visual Localization on 3D Scene Graphs"
_ICML.cc/2026/Conference — ICML 2026 regular_

### Official Review · Reviewer_PBzK · 2026-03-10

**Soundness:** 2
**Presentation:** 3
**Significance:** 2
**Originality:** 2
**Overall Recommendation:** 4
**Confidence:** 4

**Summary:**

The paper introduces SG2Loc, a lightweight sequential visual localization system utilizing 3D scene graphs and a particle filter framework. Instead of relying on large image databases or dense point clouds, the environment is represented compactly using object nodes with semantic embeddings and coarse meshes. For incoming image sequences, the method extracts per-patch semantic features and updates particle weights by comparing these features against the projected scene graph objects. To enhance accuracy, the system integrates photometric and depth supervision signals , alongside a coarse-to-fine optimization strategy and adaptive particle resampling based on KLD-sampling.

**Compliance With Llm Reviewing Policy:**

Affirmed.

**Final Justification:**

The authors have addressed the majority of my concerns; therefore, I have decided to slightly increase my score.

**Key Questions For Authors:**

1. Could you clarify the mathematical discrepancy regarding the patch extraction? Section 3.1 mentions generating 196 patches via a $14 \times 14$ grid , while Section 3.3 refers to summing similarities across 144 patches.
2. The parameters $\lambda_1$ and $\lambda_2$, which balance the depth and color scores, are set empirically. Did you experiment with learning these parameters dynamically (e.g., via an attention mechanism or end-to-end training), and how sensitive is the final pose estimation to the precise tuning of these fixed weights?

**Limitations:**

The authors adequately address the higher computational cost required for per-frame integration during raycasting and note the system's reliance on keyframing for real-time applications. However, the framework's inherent dependence on an external SLAM system for continuous ego-motion tracking  and the assumption of pre-existing, accurately segmented 3D scene graphs in the database are practical limitations that restrict its deployment in completely unmapped or highly dynamic zero-shot environments. Discussing the failure modes when the initial SLAM tracking severely diverges would provide a more complete picture of the system's limitations.

**Strengths And Weaknesses:**

Strengths:
- Significant Memory Efficiency: The proposed method achieves highly competitive localization performance while drastically reducing storage requirements. For instance, SG2Loc requires only 9.8 MB on the 3RScan dataset, compared to 294.4 MB for HLoc and 1433.6 MB for MeshLoc. This makes it highly practical for memory-constrained robotics and AR applications.
- Comprehensive Experimental Validation: The authors provide detailed ablation studies validating the contributions of individual supervision signals (semantic, depth, RGB) and the impact of varying sequence lengths. The addition of the large-scale environment simulation in the appendix further demonstrates the system's robustness.

Weaknesses:
- Limited Machine Learning Novelty (Venue Mismatch): The core contribution is primarily a systems-level integration of an existing pre-trained encoder (SceneGraphLoc) with a classic particle filter mathematical framework. The paper lacks fundamental algorithmic innovations in representation learning, neural architecture design, or theoretical optimization bounds. While an excellent contribution to robotics or computer vision systems, it is arguably less aligned with ICML's primary focus on foundational machine learning advancements.
- Empirical Hyperparameter Selection: The weighting factors for the combined multi-modal score ($\lambda_1$ and $\lambda_2$) are set empirically and kept fixed across all experiments. A learning-based approach to weigh these supervision signals dynamically would be more theoretically sound and robust.
- Mathematical/Typographical Error: There is a discrepancy in the methodology. Section 3.1 states the image is divided into a $14 \times 14$ grid, yielding 196 patches , but Section 3.3 calculates similarities by "summing these valid similarities across all 144 patches".

---

> ### Author Rebuttal · Authors · 2026-03-30
>
> **Strengths.** We thank the reviewer for highlighting the "significant memory efficiency" and "comprehensive experimental validation", and for finding the ablation studies detailed and the large-scale experiment demonstrating robustness.
>
> **W1: Limited ML novelty (venue mismatch).** We believe our work fits within ICML's scope for *application-driven machine learning*. Our contribution is formulating visual localization as a particle filtering problem on 3D scene graphs, a new problem formulation that has not been explored before. This enables localization with 1-2 orders of magnitude less storage than existing methods while achieving competitive accuracy. We also design a novel observation model that combines semantic, geometric, and photometric cues through raycasting on coarse meshes.
>
> **W2/Q2: Empirical hyperparameters lambda_1, lambda_2: learning-based weighting would be better.**
> For all experiments on both 3RScan and ScanNet, all losses are weighted equally (lambda_1 = lambda_2 = 1) and kept fixed. The fact that a single set of weights works across two datasets with different characteristics shows these values generalize without environment-specific tuning.
>
> **Loss weight ablation.** We thank the reviewer for the question. The experiment below ablates the weighting of the three observation model losses (Eq. 2) by varying lambda for semantic, depth and RGB losses. The tested scene is the first scene from the 3RScan dataset (scene 0988ea72-eb32-2e61-8344-99e2283c2728), sequence length is 5 and no PnP+Ransac refinement is applied. On this scene, equal weighting achieves the best overall performance, confirming our default choice. We will add it to the paper.
>
> | Semantic/Depth/RGB 	| Med. pos. (m) | Med. rot. (°) | Recall\@0.25m, 2°|
> |-----------------|---------------|---------------|------------|
> | **1.0/1.0/1.0** | **0.14**      	| **5.85**      	| **18.3**  |
> | **1.5**/0.75/0.75   | **0.14**  | 6.14      	| 17.3   	|
> | 0.75/**1.5**/0.75   | 0.17      	| 6.11      	| 14.4   	|
> | 0.75/0.75/**1.5**   | 0.18      	| 6.54      	| 15.4   	|
>
>
> A learning-based weighting scheme is an interesting direction for future work, but we note that keeping the weights fixed and equal is the simplest choice and already works well. We will emphasize this in the final version.
>
> **W3/Q1: 144 vs 196 patch discrepancy.**
> Thank you for catching this. This is a typo, it should be 196 patches everywhere, consistent with the 14 × 14 grid described in Section 3.1. We will fix this in the final paper.
>
> **Limitations: SLAM dependence and failure modes.**
> We agree these are relevant practical considerations. Regarding SLAM divergence: we have run an experiment using DROID-SLAM ego-motion below. It shows the results when using DROID-SLAM for relative poses both in the HLoc sequential baseline and in our SG2Loc on ScanNet for sequences of length 5. The performance remains comparable, showing that SG2Loc is robust to moderate drift in the motion estimate, while HLoc degrades more.
>
> | Method                    | Median Pos (m) &#8595; | Median Rot (°) &#8595; | Recall\@0.25m, 2° &#8593; |
> |---------------------------|----------------|----------------|------------------|
> | HLoc                      | 0.09          | 2.63          | 0.38          |
> | HLoc w/ SLAM poses        | 0.15          | 2.67          | 0.34          |
> | SG2Loc (Ours)             | 0.09          | 2.37          | 0.41          |
> | SG2Loc w/ SLAM poses      | 0.13          | 2.63          | 0.41          |
>
> Regarding the assumption of pre-existing scene graphs: our method is not tied to a specific construction method, any approach that produces object-level 3D segmentations works, including recent real-time methods like CLIO [1].
>
> [1] Maggio et al., CLIO: Real-Time Task-Driven Open-Set 3D Scene Graphs, IEEE Robotics and Automation Letters, 2024

---

> > ### Author Rebuttal · Reviewer_PBzK · 2026-04-04
> >
> > I would like to thank the authors for their clarifications and the additional experiments, which have addressed most of my concerns. Consequently, I have decided to slightly increase my score and will also take the other reviewers' comments into account during the subsequent discussion.

---

> > > ### Author Response · Authors · 2026-04-04
> > >
> > > We thank the reviewer for the feedback and for reconsidering the score. We are glad our responses addressed the concerns.

---

### Official Review · Reviewer_RGkk · 2026-03-11

**Soundness:** 3
**Presentation:** 2
**Significance:** 2
**Originality:** 3
**Overall Recommendation:** 4
**Confidence:** 3

**Summary:**

This paper proposes SG2Loc, an indoor sequential visual localization framework that replaces memory-heavy image databases and dense point clouds with lightweight 3D scene graphs. The authors frame 4-DoF pose estimation as a particle filtering problem. For a given image sequence, the system extracts patch-level semantic embeddings and uses cosine similarity to compare them against coarse 3D object meshes projected by the particles. Because semantic matches can be ambiguous, the observation model is reinforced with SSIM (photometric) and L1 depth (geometric) consistency checks. The filter's efficiency is then optimized via a coarse-to-fine downsampling strategy and adaptive KLD-sampling. Finally, a RANSAC-based PnP step refines the continuous pose by matching the query image against synthetically rendered mesh views.

The main takeaway is extreme memory efficiency. By dropping dense maps, SG2Loc cuts storage requirements by 1 to 2 orders of magnitude compared to baselines like HLoc and MeshLoc, while still achieving competitive trajectory accuracy across the 3RScan and ScanNet benchmarks. As a secondary contribution, the authors extend the original SceneGraphLoc architecture to process multi-frame sequences, which measurably improves its macro-level scene retrieval performance.

**Compliance With Llm Reviewing Policy:**

Affirmed.

**Final Justification:**

The rebuttal effectively addressed my primary methodological and practical concerns. The core contribution "memory-efficient visual localization via 3D scene graphs" is original.

My evaluation has improved based on the following resolutions:

- Runtime & Integration (Resolved): The authors clarified that SG2Loc computes absolute global poses while relying on a SLAM frontend strictly for relative ego-motion. This architectural decoupling resolves my concerns regarding delayed observation fusion and trajectory discontinuities, validating its real-world viability.

- Hyperparameter Rigor (Resolved): The provided ablation study justifies the fixed observation weights, demonstrating robust generalizability across different datasets without requiring environment-specific tuning.

- Structural Clarity (Resolved): The distinct role of sequential scene retrieval as an independent, global initialization step prior to particle filtering is now clear.

I recommend acceptance, provided the authors incorporate the following into the camera-ready manuscript:
- The explicit decoupling mechanism between the SLAM frontend and the SG2Loc backend.
- The observation weight ablation study.
- A "Societal Impact" section discussing privacy and dual-use concerns.

**Key Questions For Authors:**

The most pressing concern regarding the significance of the work is the severe computational bottleneck reported at 2.9 seconds per frame. (Q1) Could the authors provide a more granular breakdown of this inference time? Specifically, how much of this overhead is dominated by the mesh rendering and structural similarity (SSIM) or L1 depth computations for the thousands of particles  versus the semantic raycasting?

Regarding the presentation and methodological rigor of the observation model, the weights $\lambda_1$ and $\lambda_2$ used to combine the semantic, depth, and color scores in Equation 2 are explicitly stated to be set empirically and kept fixed. (Q2) Can the authors provide a quantitative sensitivity analysis for these parameters? I am particularly interested in knowing whether these fixed values generalize seamlessly across fundamentally different datasets like 3RScan and ScanNet, or if they secretly require environment-specific tuning.

Finally, the introduction of Sequential Scene Retrieval in Section 4 feels structurally disconnected from the primary pose tracking narrative. (Q3) In a practical deployment pipeline, is this sequential retrieval intended to act as a mandatory global initialization step before the particle filter begins its local tracking?

**Limitations:**

No, the authors have partially addressed the limitations but omitted the discussion on potential negative societal impacts.

The authors adequately discussed technical limitations, specifically noting the higher computational cost during per-frame integration and the method's failure to converge to precise poses when input query sequences lack discriminative information, such as when views are visually similar with little perspective change. However, the paper entirely lacks a discussion on broader societal implications.

The authors should add a brief "Societal Impact" section. For indoor visual localization technologies, relevant discussion points include:

- Privacy Concerns: The implications of algorithms that map and process representations of private or sensitive indoor environments.
- Dual-Use Potential: The ethical considerations of enabling robust autonomous navigation in GPS-denied environments, which could be utilized for surveillance or military applications.

**Strengths And Weaknesses:**

The paper's main strength lies in its original representation of the environment. Formulating sequential visual localization as a particle filtering problem on 3D scene graphs is a creative pivot away from standard dense point clouds and large image databases. Methodologically, the approach is solid. The particle filter's prediction and observation models are well-formalized , and integrating semantic, depth, and photometric cues provides a robust likelihood framework. The authors also provide thorough ablation studies that successfully isolate the impact of these multi-modal signals.

However, the method's practical significance is severely bottlenecked by its runtime. While the spatial memory reduction is significant—9.8 MB on 3RScan versus 294.4 MB for HLoc —this comes at a steep computational cost. The 2.9 seconds per frame localization time  makes real-time deployment for autonomous navigation highly impractical. On the presentation side, the manuscript needs structural work. Section 4 abruptly introduces the sequential scene retrieval task, which disrupts the main localization narrative and feels structurally disconnected. Furthermore, key methodological choices lack rigorous justification; for instance, the hyperparameters $\lambda_1$ and $\lambda_2$ used to balance the depth and color scores are fixed empirically without any theoretical grounding or sensitivity analysis.

---

> ### Author Rebuttal · Authors · 2026-03-30
>
> **Strengths.** We thank the reviewer for finding our formulation "creative", "original", the methodology "solid", the particle filter "well-formalized", and the ablation studies "thorough".
>
> **W1/Q1: Runtime of 2.9 seconds per frame is impractical.** The 2.9s per frame reflects the full coarse-to-fine particle filter, which runs three sequential passes at increasing resolutions (20, 8, 4). Importantly, this cost is per localization query, not per input frame. In practice, a SLAM system continuously tracks the camera at real-time rates (30+ fps), and SG2Loc only needs to run on sparse keyframes to correct drift and provide global pose estimates. Between keyframes, the SLAM trajectory remains valid.
> The keyframing experiment in the main paper (Table 5) validates this directly: we select keyframes at a rate matched to our processing time (effectively making our method real time), and run HLoc on the same keyframes to compare. Under this realistic setting, SG2Loc outperforms HLoc across all metrics (0.18m vs 0.27m median position error, 0.40 vs 0.35 joint recall), demonstrating that our method is already practical for robot localization when combined with a lightweight SLAM front-end.
>
>
> **W2/Q2: Hyperparameters lambda_1, lambda_2 lack justification and sensitivity analysis.**
> For all experiments on both 3RScan and ScanNet, all losses are weighted equally (lambda_1 = lambda_2 = 1) and kept fixed. The fact that a single set of weights works well across two different datasets suggests these values generalize and do not require environment-specific tuning. We will clarify this in the final version.
>
>
>
>
> **Loss weight ablation.** We thank the reviewer for the question. The experiment below ablates the weighting of the three observation model losses (Eq. 2) by varying lambda for semantic, depth and RGB losses. The tested scene is the first scene from the 3RScan dataset (scene 0988ea72-eb32-2e61-8344-99e2283c2728), sequence length is 5 and no PnP+Ransac refinement is applied.
> On this scene, equal weighting achieves the best overall performance, confirming our default choice. We will add it to the paper.
>
>
> | Semantic/Depth/RGB 	| Med. pos. (m) | Med. rot. (°) | Recall@25cm, 2° |
> |-----------------|---------------|---------------|------------|
> | **1.0/1.0/1.0** | **0.14**      	| **5.85**      	| **18.3**  |
> | **1.5**/0.75/0.75   | **0.14**  | 6.14      	| 17.3   	|
> | 0.75/**1.5**/0.75   | 0.17      	| 6.11      	| 14.4   	|
> | 0.75/0.75/**1.5**   | 0.18      	| 6.54      	| 15.4   	|
>
>
> **W3/Q3: Section 4 (Sequential Scene Retrieval) feels disconnected.** The particle filter (Section 3) and sequential scene retrieval (Section 4) solve different problems. The particle filter estimates a camera pose within a known scene. HLoc does the same as its first step. Sequential scene retrieval identifies which scene the agent is in from a database of maps. In a multi-scene deployment, retrieval would first select the correct scene, and then the particle filter would localize within it. However, the two are independent, the particle filter works without retrieval, we just uniformly sample the space. We will clarify this relationship in the revised paper.

---

> > ### Author Rebuttal · Reviewer_RGkk · 2026-04-03
> >
> > Runtime & SLAM Integration (Q1): The authors argue that a 2.9-second per-frame execution time is practical because SG2Loc only operates on sparse keyframes, while the SLAM frontend maintains the trajectory in between. This justification overlooks a fundamental issue in real-time state estimation: delayed observation fusion. At a standard 30 fps input, a 2.9-second computation time implies the SLAM frontend operates uncorrected for approximately 87 frames. The manuscript lacks any mathematical or architectural formulation (e.g., Pose Graph Optimization, factor graph formulation, or sliding window marginalization) explaining how this severely delayed global pose estimate is fused back into the real-time SLAM trajectory. Without a rigorous backend fusion strategy, injecting a nearly 3-second delayed correction will inevitably cause catastrophic trajectory discontinuities. The authors must explicitly articulate the delayed fusion mechanism to validate the claim of real-time practicality.

---

> > > ### Author Response · Authors · 2026-04-03
> > >
> > > We thank the reviewer for the follow-up. SG2Loc does not need to feed any corrections back into the SLAM trajectory. The SLAM system is only used to get relative ego-motion between frames, this is the motion model for our particle filter. The particle filter runs on top of that and directly estimates the global pose in the map's coordinate frame. That is our localization output. There is no conflict between two global estimates that would need to be reconciled. SLAM gives us relative motion, SG2Loc gives us an absolute pose.
> > >
> > > Apart from that, SG2Loc can be used as a map-based relocalization module to correct SLAM drift. Systems like maplab 2.0 (Cramariuc et al., 2022) are explicitly designed to fuse absolute 6-DoF pose estimates from external localization systems as factors in their pose graph optimization. Corrections are applied whenever a localization result is available, not continuously. SG2Loc provides exactly this, an absolute pose in the map's coordinate frame, and could be integrated as an external localization source in such a framework. We will clarify this in the paper.

---

### Official Review · Reviewer_rZxr · 2026-03-12

**Soundness:** 2
**Presentation:** 3
**Significance:** 2
**Originality:** 2
**Overall Recommendation:** 4
**Confidence:** 3

**Summary:**

This paper is about visual localization from query images. The authors propose to encode the scene into a coarse 3D mesh and a scene graph that holds object features. The proposed algorithm is based on particle filtering. Given a coarse ego-pose, the 3D mesh is projected into the image. Image patches are then compared with the objects from the scene graph via their encoded features. A comparison via depth sensors is possible, too. In a last step, a pose refinement is done by sampling 6 synthetic images around the pose discovered in the previous stage and selecting the pose of the best matching image.

**Compliance With Llm Reviewing Policy:**

Affirmed.

**Final Justification:**

I thank the authors for their comments and explanations. While I partially agree with some of the other reviewers regarding the technical novelty of this paper, I also acknowledge the authors' point of view that this is a novel combination of algorithms which might be interesting to the SLAM community. I will slightly raise my score.

**Key Questions For Authors:**

- Can the algorithm work without known ego-motion?

- Why is the performance on ScanNet worse than on 3RScan?

**Limitations:**

yes

**Strengths And Weaknesses:**

- The paper is clearly written, and easy to read.

- If a coarse ego-pose is required to run the algorithm, the applicability is much more limited.

- From table 1, we can see that for ScanNet, the proposed algorithm is competitive with SOTA only for sequences of length 5. For longer sequences, SOTA is superior. Why does the algorithm not work as well on ScanNet as on 3RScan?

---

> ### Author Rebuttal · Authors · 2026-03-30
>
> We thank the reviewer for finding the paper "clearly written" and "easy to read".
>
> **W1: Coarse ego-pose required, limiting applicability.**
> Our method does not require a coarse absolute pose, it only needs relative motion between consecutive frames, which any visual odometry or SLAM system provides (e.g., DROID-SLAM, ORB-SLAM). These are standard components readily available on robots, phones, and AR headsets. The absolute pose is what our method estimates.
>
>
> **W2: On ScanNet, competitive only for short sequences. SOTA is superior for longer ones. && Q2: Why is performance on ScanNet worse than on 3RScan?**
> We respectfully note that SG2Loc's absolute performance is actually comparable across both datasets, for example at 25 frames, median errors are 0.07m/2.21° on 3RScan and 0.08m/2.17° on ScanNet. On 3RScan, SG2Loc achieves the best joint recall for all sequence lengths. On ScanNet, SG2Loc achieves the best joint recall for 5 and 10 frames and second-best for 25 frames (0.46 vs HLoc 0.47). So the method is not only competitive for short sequences, it performs well across all lengths on both datasets. The small gap at 25 frames on ScanNet comes with a 10x storage reduction: 28 MB versus HLoc 283 MB (Table 2).
>
> **Q1: Can the algorithm work without known ego-motion?**
> Our method is a particle filter and thus requires a motion model to propagate particles between frames. In practice, ego-motion can be estimated by any SLAM or visual-inertial odometry (VIO) system. Any device that captures image sequences already provides ego-motion estimates (iPhones, Android phones, AR headsets, and robots all have built-in VO capabilities). We will add the experiment below using DROID-SLAM ego-motion to the final paper. It shows the results when using DROID-SLAM for relative poses both in the HLoc sequential baseline and in our SG2Loc on ScanNet for sequences of length 5. The performance remains comparable, showing that SG2Loc is robust to moderate drift in the motion estimate, while HLoc degrades more.
>
> | Method                    | Median Pos (m) &#8595; | Median Rot (°) &#8595; | Recall\@0.25m, 2° &#8593; |
> |---------------------------|----------------|----------------|------------------|
> | HLoc                      | 0.09          | 2.63          | 0.38          |
> | HLoc w/ SLAM poses        | 0.15          | 2.67          | 0.34          |
> | SG2Loc (Ours)             | 0.09          | 2.37          | 0.41          |
> | SG2Loc w/ SLAM poses      | 0.13          | 2.63          | 0.41          |

---

> > ### Author Rebuttal · Reviewer_rZxr · 2026-04-04
> >
> > I thank the authors for their comments and explanations. While I partially agree with some of the other reviewers regarding the technical novelty of this paper, I also acknowledge the authors' point of view that this is a novel combination of algorithms which might be interesting to the SLAM community. I will slightly raise my score.

---

> > > ### Author Response · Authors · 2026-04-04
> > >
> > > We thank the reviewer for the response and for raising the score. We appreciate the acknowledgement that our algorithm offers value to the SLAM community, and we are glad our responses adequately addressed the concerns.

---

### Official Review · Reviewer_n71c · 2026-03-13

**Soundness:** 3
**Presentation:** 3
**Significance:** 2
**Originality:** 2
**Overall Recommendation:** 3
**Confidence:** 3

**Summary:**

This paper addresses indoor visual localization from image sequences. Instead of relying on storage-heavy image databases or dense 3D maps, it uses a compact 3D scene graph for lightweight sequential localization. The method is built within a particle filter framework, where each particle represents a candidate camera pose and is updated using ego-motion from SLAM together with patch-level semantic similarity between the observed image and the scene projected from that pose. Additional cues such as depth and RGB appearance can also be incorporated to improve robustness. Experiments on 3RScan and ScanNet show that the method achieves competitive localization accuracy while requiring significantly less storage than strong baselines such as HLoc and MeshLoc. The results also show that longer sequences improve performance, and ablation studies confirm that depth, RGB cues, and the final PnP refinement all contribute to the overall accuracy.

**Compliance With Llm Reviewing Policy:**

Affirmed.

**Final Justification:**

The authors’ rebuttal clarified some of my concerns. Nevertheless, I remain unconvinced about the significance of the paper’s technical contribution, and thus I maintain my weak reject score.

**Key Questions For Authors:**

As noted in the weaknesses, I feel the method is more engineering-driven and depends on the combination of several external components. For that reason, I think the paper would benefit from more detailed studies of these components and their impact on the final performance.

**Limitations:**

yes

**Strengths And Weaknesses:**

Strengths:

- The method avoids storing large image databases or dense 3D maps by using a lightweight 3D scene graph, which is a very practical design for storage-constrained and real-world deployment settings.

- The experiments demonstrate practical advantages in terms of storage efficiency and runtime. Despite relying on a much more compact scene representation than previous approaches, it still achieves good localization performance.

- The ablation study supports its design choice in combining semantic, depth, and RGB cues.


- Overall, the paper is easy to read and generally well organized. The method is presented in a clear and intuitive way, and the motivation for using scene graphs and particle filtering is easy to follow.

Weakness:

- The overall system appears more like an engineering-driven combination of existing components, and I find the methodological and technical novelty somewhat limited. While I appreciate the effort involved in designing a complete and effective pipeline, I still think this is a notable limitation of the work.


- The paper does not provide an analysis of how varying the number of particles affects the tradeoff between localization accuracy and runtime. Beyond the particle count, it also remains unclear how sensitive the method is to other hyperparameters, such as the motion noise parameters and the weighting terms used in particle score computation. A discussion or ablation on these design choices would help clarify the robustness of the method and whether its performance depends strongly on careful hyperparameter tuning.

- The paper can add more discussion on how the scene graph is constructed in practice.

---

> ### Author Rebuttal · Authors · 2026-03-30
>
> **Strengths.**
> We thank the reviewer for finding our method "lightweight", "practical design", "storage efficient", "easy to read", "well organized", and the "motivation for scene graphs and particle filtering easy to follow".
>
> **W1: Engineering-driven combination, limited novelty.**
> We respectfully disagree. While individual components (particle  filter, scene graph, semantic matching) exist separately, our core contribution is the formulation of visual localization as a particle filtering problem on 3D scene graphs. This is a new problem formulation that has not been explored before. Rather than simply plugging components together, we design a unified observation model that combines semantic, geometric, and photometric cues through raycasting on coarse meshes, replacing the dense point clouds or image databases that prior methods require. We believe this work fits well within ICML's scope of *application-driven machine learning*, where the contribution lies in showing that a principled combination of ideas can open up a new and practical direction for localization.
>
> **W2: No analysis of particle count vs. accuracy/runtime tradeoff.**
> We thank the reviewer for the suggestion. The table shows the effect of the particle number on runtime and accuracy. We ran this setting 2,000 minimum number particles / 10,000 maximum number particles for the paper as this gives the best tradeoff between runtime and accuracy. The table shows only the coarse pose obtained by the particle filter, no refinement with RANSAC and PnP:
>
> | Min / Max #Particles | Avg. #Particles | Med. pos. (m) | Med. rot. (°) | Pos. R@0.25m | Rot. R@2° | R@25cm, 2° | Avg. t/frame (s) |
> |----------------------|----------------|---------------|---------------|--------------|-----------|------------|------------------|
> | 500 / 1,000       | 746           | 0.25          | 3.72          | 49.1      | 30.2      | 25.2      | 0.97          |
> | 2,000 / 10,000    | 2221          | **0.22**      | **3.67**      | **52.8**  | 34.0      | **30.2**   | 2.60             |
> | 4,000 / 20,000    | 4113          | **0.22**      | 4.15          | 50.9      | **35.8**  | **30.2**   | 3.94             |
>
> **W2: Sensitivity to hyperparameters.**
> The motion noise parameters (sigma_trans=0.05m, sigma_rot=0.05 rad) and the likelihood sigma (0.2) were set once and kept fixed across all experiments on both 3RScan and ScanNet. The loss weights lambda_1 and lambda_2 are set equally (all losses weighted the same, all lambda are set to 1). The fact that a single set of hyperparameters works well across two different datasets suggests the method is not sensitive to careful tuning, which is a good thing we believe.
>
> **Loss weight ablation.** We thank the reviewer for the question. The experiment below ablates the weighting of the three observation model losses (Eq. 2) by varying lambda for semantic, depth and RGB losses. The tested scene is the first scene from the 3RScan dataset (scene 0988ea72), sequence length is 5 and no PnP+Ransac refinement is applied.
> On this scene, equal weighting achieves the best overall performance, confirming our default choice. We will add it to the paper.
>
> | Semantic/Depth/RGB 	| Med. pos. (m) | Med. rot. (°) | R@25cm, 2° |
> |-----------------|---------------|---------------|------------|
> | **1.0/1.0/1.0** | **0.14**      	| **5.85**      	| **18.3**  |
> | **1.5**/0.75/0.75   | **0.14**  | 6.14      	| 17.3   	|
> | 0.75/**1.5**/0.75   | 0.17      	| 6.11      	| 14.4   	|
> | 0.75/0.75/**1.5**   | 0.18      	| 6.54      	| 15.4   	|
>
> **W3: More discussion on scene graph construction.** We use existing methods to build the scene graphs. For ScanNet, we use SceneGraphFusion [1], which incrementally predicts 3D scene graphs from RGB-D sequences. For 3RScan, scene graph annotations are provided with the dataset. Our method is not tied to a specific scene graph construction method, it only requires object meshes and semantic embeddings. Any method that produces object-level 3D segmentations can be used, such as CLIO [2], which builds scene graphs from open-vocabulary features in real time. We will add this to the paper.
>
> [1] Wu et al., SceneGraphFusion: Incremental 3D Scene Graph Prediction from RGB-D Sequences, CVPR 2021
>
> [2] Maggio et al., CLIO: Real-Time Task-Driven Open-Set 3D Scene Graphs, IEEE Robotics and Automation Letters, 2024
>
> **Key Question: More detailed studies of external components.**
> We provide ablations in Tab. 7 that show the contribution of each component: the semantic score, depth score, RGB score, adaptive resampling, coarse-to-fine optimization, and PnP refinement. Each component adds measurable improvement. Tab. 8-10 in the appendix further break down the individual supervision signals.

---

> > ### Author Rebuttal · Reviewer_n71c · 2026-04-04
> >
> > I thank the authors for their rebuttal, which addressed my questions on the number of particles and hyperparameters. Still, regarding the contribution, I am not full convinced by authors’ reply. Just to clarify, I do not doubt this work lies within the ICML scope. Instead, I find the technical contribution of the proposed method is not that significant.

---

> > > ### Author Response · Authors · 2026-04-04
> > >
> > > We respectfully disagree that the technical contribution is not significant. To the best of our knowledge, SG2Loc is the first method to perform metric pose estimation directly on a 3D scene graph. All prior scene graph work is limited to coarse place retrieval. Our observation model is principled, jointly integrating semantic, photometric, and geometric cues, unlike prior particle filter methods (e.g., Loc-NeRF) that rely only on photometric error on dense representations. We achieve competitive accuracy while requiring much less storage (Table 2). This is particularly relevant for robotics, where 3D scene graphs are already used for downstream tasks like navigation, planning, and manipulation. SG2Loc enables localization on the same representation without building a separate map. We kindly ask the reviewer to specify which aspect they find insufficient so we can respond more precisely.

---

### Decision · Program_Chairs · 2026-04-30

**Decision:**

Accept (regular)

**Comment:**

After the author rebuttals, the work received 3 weak accept, 1 weak reject recommendations. ACs note that the pros and cons of the work evaluated by the reviewers are quite consistent, they however see the novelty different. While the reviewer n71c sees this as a more engineering work via the combination of existing components, RGkk notes the core contribution "memory-efficient visual localization via 3D scene graphs" is original. A number of questions initially raised by the reviewers have been well addressed by the author rebuttals. ACs recommend acceptance.